# Chimeric Antigen Receptor T Cells in Glioblastoma—Current Concepts and Promising Future

**DOI:** 10.3390/cells12131770

**Published:** 2023-07-03

**Authors:** Rebecca Kringel, Katrin Lamszus, Malte Mohme

**Affiliations:** Department of Neurosurgery, University Medical Center Hamburg-Eppendorf, 20246 Hamburg, Germany; rebecca.kringel@stud.uke.uni-hamburg.de (R.K.); lamszus@uke.de (K.L.)

**Keywords:** CAR-T cells, immunotherapy, GBM, glioma, glioblastoma, T cells

## Abstract

Glioblastoma (GBM) is a highly aggressive primary brain tumor that is largely refractory to treatment and, therefore, invariably relapses. GBM patients have a median overall survival of 15 months and, given this devastating prognosis, there is a high need for therapy improvement. One of the therapeutic approaches currently tested in GBM is chimeric antigen receptor (CAR)-T cell therapy. CAR-T cells are genetically altered T cells that are redirected to eliminate tumor cells in a highly specific manner. There are several challenges to CAR-T cell therapy in solid tumors such as GBM, including restricted trafficking and penetration of tumor tissue, a highly immunosuppressive tumor microenvironment (TME), as well as heterogeneous antigen expression and antigen loss. In addition, CAR-T cells have limitations concerning safety, toxicity, and the manufacturing process. To date, CAR-T cells directed against several target antigens in GBM including interleukin-13 receptor alpha 2 (IL-13Rα2), epidermal growth factor receptor variant III (EGFRvIII), human epidermal growth factor receptor 2 (HER2), and ephrin type-A receptor 2 (EphA2) have been tested in preclinical and clinical studies. These studies demonstrated that CAR-T cell therapy is a feasible option in GBM with at least transient responses and acceptable adverse effects. Further improvements in CAR-T cells regarding their efficacy, flexibility, and safety could render them a promising therapy option in GBM.

## 1. Introduction

Glioblastoma (GBM) is the most common malignant primary brain tumor in adults [1,2]. It represents 45.6% of primary malignant brain tumors, with an annual incidence of about 3 per 100,000 [3,4]. GBM shows a diffusely infiltrative growth and an aggressive behavior with a high degree of drug resistance [5,6]. Risk factors for the development of GBM are poorly defined, except for radiation exposure [7]. To date, GBM remains largely refractory to treatment and there is no curative option. The standard of care includes maximal safe surgical resection and adjuvant chemoradiotherapy with temozolomide. However, GBM has the capacity to escape anti-tumor therapy via multiple signaling pathways and, therefore, invariably relapses. GBM patients have a five-year cumulative survival of about 5% [1,7]. Depending on the patient’s performance status and symptoms, the treatment of recurrent GBM can be performed using second-line chemotherapy or can further include treatment with the anti-vascular endothelial growth factor (VEGF) antibody bevacizumab, alternating field therapy, or enrolment in clinical trials [1,8]. Tumor/glioma stem-like cells (GSC) are hypothesized to contribute to tumor initiation, relapse, metastasis, and poor survival in multiple malignancies, including GBM. Typically, GSCs are capable of self-renewal, proliferation, and differentiation into tumor cells with multiple expression markers [9]. In addition to the assumed mechanism of GSC-driven tumor progression, recent single-cell analyses have identified different molecular cell subtypes which can co-exist in the same tumor, leading to a high grade of heterogeneity in GBM [10,11,12]. This heterogeneity requires a more personalized approach in the treatment of GBM [3,13].

Various immunotherapeutic approaches, such as peptide vaccines, dendritic cell therapy, and immune checkpoint inhibitors, have been tested in GBM patients [14]. So far, none of these approaches has led to promising results, due to several obstacles for immunotherapy in GBM, including anatomical barriers and immunosuppression in the TME [15]. Considering the poor prognosis after standard first-line treatment and the lack of promising second-line therapies, it is important to assess the potential benefit of CAR-T cell therapy in GBM [14].

CAR-T cells have achieved groundbreaking success in hematologic malignancies, as demonstrated in several clinical trials of cluster of differentiation 19 (CD19)-CAR-T cell therapy, with complete response rates of 50–90% in refractory B-cell malignancies [16,17]. Two CAR-T cell therapies are currently approved: Tisagenlecleucel (Kymriah) is used to treat patients with acute lymphoblastic leukemia (ALL), while axicatagene ciloleucel (Yescarta) is approved for the treatment of large B-cell lymphoma, such as non-Hodgkin lymphoma [18].

CARs are genetically modified receptors that are designed to target a specific tumor antigen and redirect T lymphocytes to attack tumor cells [18]. Possible targets are lipid, protein, and carbohydrate antigens [17,19,20,21]. Cytolytic effector functions include exocytosis of cytotoxic granules containing perforin and granzyme, and the expression of membrane-bound tumor necrosis factor (TNF) family ligands, which induce target cell apoptosis after binding to their respective receptor [21]. First-generation CARs only contain a CD3ζ signaling chain to induce T cell activation after antigen binding [21,22]. Second-generation CARs additionally have a co-stimulatory signaling domain to provide a second activation signal and, therefore, show a higher therapeutical efficacy [20]. Third-generation CARs include two co-stimulatory domains with an attempt to mimic the co-stimulation that is provided during TCR recognition by antigen presenting cells (APCs) [17,18,21]. Fourth- or fifth-generation CAR-T cells include signaling domains from cytokine receptors or inducible expression of inflammatory cytokines like IL-12 or IL-18 [21]. These fourth-generation CAR-T cells are called “T-cells redirected for universal cytokine-mediated killing (TRUCK)” and are able to not only kill tumor cells via tumoricidal cytokines but also induce a second wave of immune recruitment [20].

Various parameters can influence the response to CAR-T cells: (i) the specific target antigen; (ii) the CARs’ structure; (iii) the CAR-T cell dose, frequency, and route of administration; (iv) the TME; (v) the patient’s treatment prior to CAR-T cell administration; and (vi) the engraftment and trafficking capacity of the CAR-T cells [16,19,23]. An ideal CAR-T cell target is defined by (i) a high expression on the tumor cell surface, (ii) a high degree of intra- and intertumoral homogeneity, (iii) being essential in order to prevent immune escape via antigen loss, and (iv) no expression on normal tissue to avoid on-target/off-tumor toxicity [24]. On-target/off-tumor toxicity results from interactions between the CAR and non-malignant cells that express the target antigen [25].

Many approaches to target solid tumors in at least 20 different organs with CAR-T cells have been described [26]. IL-13Rα2 was the first target tested in patients with GBM. EGFRvIII, HER2, and EphaA2 are other major targets in GBM due to their overexpression on GBM cells but lack of expression in healthy brain tissue [27]. Targets and their characteristics as well as preclinical and clinical trials of CAR-T cell therapy in GBM are summarized in Figure 1 and Appendix A.

## 2. Preclinical Studies

### 2.1. Targeting IL-13Rα2

Several preclinical studies were performed in orthotopic glioma mouse models with CD8+ CAR-T cells that targeted IL-13Rα2, including a CAR construct that was optimized with a 4-1BB co-stimulatory domain [28,29,30,31,32]. This optimization led to improved anti-tumor activity and T cell persistence compared to first-generation CARs.

Additionally, application of dexamethasone did not diminish CAR-T cell anti-tumor activity, and local intracranial delivery of CAR-T cells led to superior anti-tumor efficacy when compared to intravenous (i.v.) administration. Treated mice showed a significantly improved survival [28].

In another study performed in an orthotopic GBM mouse model, CD4+ CAR-T cells targeting IL-13Rα2 were tested and compared to CD8+ CAR-T cells. While CD8+ cells exhibited a robust short-term effector function but became rapidly exhausted, CD4+ CAR-T cells persisted after tumor interaction and sustained their effector potency. CD4+ CAR-T cells were equally able to mediate cytotoxic effects against GBM via degranulation-mediated mechanisms as in CD8+ T cells. Interestingly, the mixing of both types of cells led to the impairment of CD4+ CAR-T cell effector potency. CD4+ CAR-T cells outperformed CD8+ CAR-T cells, especially regarding long-term tumor eradication and survival of treated mice [29].

### 2.2. Targeting EGFR and EGFRvIII

A third-generation EGFRvIII CAR-T cell with a CD28 and a 4-1BB co-stimulatory domain was evaluated in a syngeneic, fully immune-competent mouse model after challenge with the murine glioma parental cell lines SMA560vII. In this approach, intravenous (i.v.) CAR-T cell infusion with elevated doses was able to cure all mice. Cured mice were resistant to rechallenge with EGFRvIII negative tumors, suggesting that host immunity against additional antigens could have been generated [33].

Another approach included synthetic Notch (synNotch) CAR-T cells. In synNotch CAR-T cells, the binding of a first antigen with the synNotch receptor induces the transcription of a second CAR that can recognize a different antigen on cancer cells. Therefore, the overexpression of both antigens is required to enable CAR-T cell activation and effector function [27]. This technique prevents tonic signaling and exhaustion and, thus, leads to a higher fraction of CAR-T cells in a memory state. An orthotopic mouse model bearing patient-derived xenograft (PDXs) GBM was treated with a single i.v. infusion of EGFRvIII synNotch CAR-T cells and experienced a complete and long-term remission as well as an increased survival compared to constitutive CARs [34].

Bispecific CARs are able to recognize two different antigens on two distinct antigen recognition domains, which leads to a synergistic cascade of effector molecules. Additionally, bispecific CARs can conquer tumor evasion in case of antigen mutation or loss of one of the antigens [16,18]. Intravenously applied CAR-T cells, that were designed to target an epitope of EGFRvIII and a form of EGFR expressed on tumor cells but not on EGFR-expressing normal cells, were able to effectively lyse EGFRvIII- or EGFR-overexpressing tumor cells in a subcutaneous xenograft mouse model injected with EGFR-expressing tumor cells of the U87 and U251 cell lines. This effect could be enhanced by incorporating a 4-1BB co-stimulatory domain. Treated mice showed a significantly prolonged survival [35].

Furthermore, CAR-T cells can be engineered to secrete bispecific T cell engagers (BiTes). BiTes are able to bring T cells and target antigens on cancer closely together by binding both of them. This leads to an activation of downstream killing pathways in the corresponding T cell [36]. Intraventricular injection with CAR-T-EGFRvIII.BiTE-EGFR cells not only induced complete and durable responses associated with prolonged survival in an orthotopic mouse model that received tumor cells of the human glioma U87 and U251 cell lines, as well as in PDXs, but also redirected non-specific bystander T cells and Tregs to become cytotoxic killers without on-target/off-tumor activity [37,38].

Another group combined EGFRvIII-CAR-T cells with intratumoral IL-12 delivery. IL-12 is a pro-inflammatory cytokine which can directly support the cytotoxic activity of T cells as well as improve antigen presentation and reshape inhibitory cells within the TME. In an immunocompetent orthotopic GBM model, which received EGFRvIII positive GL261 cells intravenously, EGFRvIII CAR-T cells alone failed to control fully established tumors. In combination with a single locally delivered dose of IL-12, a durable anti-tumor response and long-term survival in about 50% of treated mice could be achieved [39].

In order to improve the delivery and efficacy of CAR-T cells, anti-VEGF therapy was combined with i.v. EGFRvIII CAR-T cells in immunocompetent mice bearing orthotopic GBM tumors from CTA2 and GSC500 cell lines that were engineered to express EGFRvIII. The combinational approach led to improved CAR-T cell infiltration, delayed tumor growth, and prolonged survival of treated mice compared to EGFRvIII CAR-T cells alone [40].

### 2.3. Targeting HER2 and Combinatorial Targets

A strategy to limit the risk of on-target/off-tumor toxicity and conquer antigen escape is to use Tandem CAR-T (TanCAR-T) cells. TanCAR-T cells join the single-chain variable fragments of two different antigens, which can lead to an enhanced T cell activation after binding both antigens simultaneously. This approach was tested for targeting HER2 and IL-13Rα2 in an orthotopic xenogeneic GBM mouse model with the use of intratumoral injections: TanCAR-T cells showed enhanced anti-tumor efficacy and improved animal survival [30]. In addition, TanCAR-redirected T cells targeting IL-13Rα2 or EphA2 in a xenograft mouse model, after subcutaneous injection of U87 glioma cells, were able to elicit greater tumor regression than single CAR-T cells with the potential for reduction of antigen escape and off-target cytotoxicity [41].

Trivalent CAR-T cells co-targeting HER2, IL-13Rα2, and EphA2 could overcome interpatient variability, achieve a capture of antigens in nearly 100% of tumor cells, avoid antigen escape, and were shown to increase the survival in an orthotopic xenograft model of GBM. Furthermore, trivalent CAR-T cells are thought to have enhanced signaling and ability to mediate a robust immune synapse [31].

### 2.4. Targeting EphA2

An et al. studied third-generation EphA2 CAR-T cells with CD28 and 4-1BB as co-stimulatory domains applied intravenously in a xenograft mouse model with a subcutaneous injection of U251 glioma. They found that the anti-tumor activity of the CAR-T cells was related to the upregulation of CXCR-1/2 and appropriate IFNγ production. CAR-T cells with a high level of IFNγ showed poor anti-tumor activity due to upregulation of PD-L1 in GBM cells. Accordingly, the PD1-blockade improved the efficacy of CAR-T cells with poor anti-tumor activity [42]. However, after loss of the genes in the IFNγ receptor signaling pathway, GBM and other solid tumors were more resistant to killing by CAR-T cells due to a lower upregulation of cell-adhesion pathways and, subsequently, reduced overall CAR-T cell binding duration and avidity. These findings underline the dual role of IFNγ in modulating CAR-T cell effector functions [43].

### 2.5. Targeting NKG2D Ligands (NKG2DLs)

The NKG2DLs are expressed on some normal cells, but can be upregulated on infected or tumor cells. The respective receptor is found on natural killer (NK) cells as well as CD8+ T cells [44]. NKG2D CAR-T cells lysed NKG2DL-expressing GBM cells and GSCs in vitro and eliminated xenograft tumors of the U251 and U87 cell lines in vivo, without significant treatment-related toxicity. The advantages of CAR-T cells targeting NKG2DLs are their ability to (i) lyse immunosuppressive cells, such as myeloid-derived suppressor cells (MDSCs) and Tregs; (ii) eliminate neovasculates in the TME; (iii) target multiple tumor-associated ligands, which may prevent immune escape caused by the tumors’ heterogeneity; and (iv) their improved persistence [45].

Another study in an orthotopic GBM mouse model implanted with GL-261 cells revealed the synergistic effect of NKG2D CAR-T cells and radiotherapy (RTx) with enhanced migration of the CAR-T cells to the tumor site after i.v. application and increased effector functions [46].

### 2.6. Targeting GD2

The GD2 CAR-T cells led to a significantly improved survival rate in mice bearing GBM when injected intracerebrally, but not when administered intravenously [47]. In another approach, GD2-targeting CAR-T cells with additional transgenic IL-15 expression showed an improved tumor control and survival of intravenously treated mice bearing patient-derived intracranial GBM [48]. Treatment with GD2 CAR-T cells in combination with the dual IGF1R/IR antagonist, Linsitinib, decreased the activation/exhaustion of GD2 CAR-T cells and increased their central memory profile and anti-tumor activity in an orthotopic xenograft mouse model, challenged with primary patient-derived cell lines of GBM [47,48,49].

### 2.7. Other Targets

The B7-H3 (CD276) CAR-T cells significantly increased the median survival in an orthotopic GBM mouse model [50]. Another study showed that B7H3-targeted CAR-T cells in combination with intratumoral administration of a CXCL 11-armed oncolytic adenovirus could achieve a durable antitumor response. The immunosuppressive TME was altered, with increased infiltration of CD8+ T cells, NK cells, and M1-polarized macrophages, whereas the proportions of MDSCs, Tregs, and M2-polarized macrophages were reduced [51].

The CD70-specific CAR-T cells demonstrated a significant antitumor activity in different murine AML xenograft models [52]. In GBM mouse models, CD70 CAR-T cells were able to generate a potent anti-tumor response and improved survival in treated mice without adverse events [53,54]. A combinatorial approach with CD70 CAR-T cells and an oncolytic herpes simplex virus-1 (oHSV-1) led to an enhanced pro-inflammatory environment, including an increased proportion of T cells with enhanced activity and a reduced number of Tregs in the TME [53,54,55].

The CD133 CAR-T cells showed efficient anti-tumor effects in a xenograft mouse model, without causing adverse effects on normal CD133+ hematopoietic stem cells. After intracranial injection, CAR-T cells persisted and stayed localized to tumor areas. Therefore, a combination of standard of care with CD133 CAR-T cells could be another feasible option [56].

## 3. Clinical Studies

Some of the previously explored targets in preclinical studies were translated into clinical trials.

### 3.1. Targeting IL-13Rα2

Intracranial delivery of IL-13Rα2-directed CAR-T cells into the resection cavity of three patients with recurrent GBM led to a transient anti-tumor response in two patients, as indicated by an increased MRI gadolinium enhancement and increased FLAIR signal [57]. An analysis of tumor tissue from one patient with a second recurrence revealed a reduced overall IL-13Rα expression within the tumor after treatment. Grade 3 or higher adverse events, including transient leukopenia, headaches, and fatigue, only occurred after administration of the highest dose, indicating an acceptable safety profile of the therapy. Nevertheless, GBM recurred in all treated patients, most likely due to antigen loss [57]. Another GBM patient with intracranial and spinal tumor manifestations received multiple intralesional infusions followed by intraventricular infusions. After application of CAR-T cell treatment, regression of all intracranial and spinal lesions could be observed. The clinical response continued for 7.5 months after initiation of CAR-T cell therapy with no toxic effects of grade 3 or higher. Grade 1 or 2 adverse effects included headaches, generalized fatigue, myalgia, and olfactory auras. Preliminary results indicated decreased expression of IL-13Rα2 after tumor recurrence [58]. 

Brown et al. tested off the shelf, healthy donor-derived, steroid-resistant IL-13Rα2-targeted CAR-T cells which were infused locally into the tumor via a Rickham catheter, in combination with intracranial administration of recombinant human IL-2 (aldesleukin) and systemic dexamethasone in a phase I trial in six patients with recurrent GBM. The authors avoided Graft-versus-Host Disease by selecting an oligoclonal T cell population that was evaluated prior to clinical administration for its alloreactivity. As a result, the therapy was well tolerated and led to a local transient effect. Nevertheless, all tumors eventually recurred, and the T cells did not persist in the TME. In a still ongoing study, IL-13Rα2-targeted CAR-T cells with a 4-1BB costimulatory domain achieved improved antitumor activity and persistence with evidence of clinical activity in a subset of patients, including one patient who displayed a complete response [59].

### 3.2. Targeting EGFRvIII

O’Rourke et al. studied CAR-T cells directed against EGFRvIII that were intravenously infused into 10 patients with recurrent GBM. The median OS was about eight months, and one patient has had a residual stable disease of over 18 months of follow-up. EGFRvIII CAR-T cells were able to traffic to regions of active GBM. Comparisons of EGFRvIII-expression from tumor tissue resected before and after infusion showed an antigen decrease in five of seven patients. Regarding safety, no off-tumor toxicity or cytokine release syndrome (CRS) were observed [60].

An approach with a third-generation EGFRvIII-CAR construct with a CD28 and 4-1BB co-stimulatory domain and IL-2 application post transfer, both administered intravenously administered in 18 patients, led to a median OS of 6.9 months, with two patients surviving more than one year and a third patient being alive at 59 months. Two patients experienced severe hypoxia, including one treatment-related mortality, most likely due to congestion of the pulmonary vasculature from activated T cells and consecutive pulmonary edema. Since most patients demonstrated progressive disease with a median progression-free survival of 1.3 months, the authors concluded that the approach was not capable of inducing objective tumor regression and failed in prolonging survival in patients with recurrent GBM [61].

### 3.3. Targeting HER2

HER2-CMV-bispecific CAR-T cells were tested in a phase I trial. By expression of CARs in virus-specific T cells, CAR-T cells provided the expected anti-tumor activity via their CAR binding to the tumor-associated antigen (TAA), but also received co-stimulation by latent virus antigens presented by APCs after native T cell receptor engagement. In the phase I trial, half of the 16 treated patients had progressive disease while the other half had an objective response after administration of HER2-CMV-CAR-T cells. The median OS was 11.6 months from infusion and 24.8 months from diagnosis: 38% of patients experienced a durable clinical benefit [62].

### 3.4. Targeting EphA2

A first-in-human trial of EphA2-redirected CAR-T cells was performed in three patients with recurrent glioblastoma. The patients received a single dose of intravenously applied CAR-T cells. Two patients suffered from grade 2 CRS, accompanied by pulmonary edema, which could be resolved completely with dexamethasone medication. Apart from that, no other organ toxicity including neurotoxicity was observed. After infusion, CAR-T cells expanded and persisted for more than four weeks, and a transient size reduction of the tumor was observed. This response was sustained for less than one month, paralleled by the persistence of CAR-T cells [63].

### 3.5. Targeting GD2

GD2 CAR-T cells administered intraventricularly in four patients with H3K27M-mutated diffuse midline gliomas led to a clinical and radiographic improvement with reversible toxicity in the tumor area. There was less systemic toxicity, such as CRS, compared to the i.v. infusions [64]

In another study that included eight patients with progressive GMB, GD2-specific fourth-generation safety-designed CAR-T cells with an inducible suicide caspase 9 gene were tested either i.v. or i.v. combined with intracavitary infusions. The treatment was well tolerated, and the CAR-T cells were able to partially mediate antigen loss. Consequently, the lifespan was extended in some patients, although the clinical benefit could not be determined in this study due to the small cohort size [65].

## 4. Challenges

Currently, five main categories of immunotherapies are being studied in clinical trials in GBM—vaccines, cytokine therapy, oncolytic viral therapy, checkpoint inhibitors, and CAR-T therapy—as well as their combination. To date, no immunotherapy has demonstrated a survival benefit in GBM over standard of care [8]. In particular, patients with GBM that received CAR-T cell therapy did not gain a clinical benefit in contrast to the complete remissions observed in hematological malignancies. There are several hurdles set up by GBM for CAR-T cell therapy: (i) neuroanatomical barriers such as the blood–brain barrier (BBB)/blood–brain tumor barrier (BBTB) [5,66]; (ii) genetic and molecular characteristics such as intra- and intertumoral heterogeneity [5,67] as well as (iii) low mutational burden and lack of stably expressed clonal antigens [7]; (iv) immune evasion due to the highly immunosuppressive TME [8,66], additionally to (v) a systemic immunosuppression in GBM patients [7]; and last but not least, (vi) a limited understanding of the basic cellular mechanisms leading to resistance to anti-tumor therapies [66]. On the other hand, there are several challenges that arise from the usage of CAR-T cells themselves: (i) toxicity, including systemic reactions such as CRS as well as on-target/off-tumor toxicity; (ii) production difficulties of autologous CAR-T cell manufacturing; and (iii) suboptimal persistence in vivo [19,25].

The limitations of GBM mouse models can also lead to divergent results between preclinical and clinical studies. The main pitfalls of in vivo models of GBM include deviating biological properties and the lack of cellular heterogeneity compared to patient tumors, as well as different pharmacokinetics in animals compared to humans. Especially tumor cells derived from GBM cell lines experience a genetic drift leading to homogeneous cell populations. While PDXs can more accurately replicate human GBM regarding intratumoral heterogeneity, the standardization and reproducibility of the experimental results are limited. Furthermore, xenograft models of GBM—such as cell lines derived from human tissue and PDXs—require immunodeficient mice, limiting their ability to accurately reflect the tumor-immune interaction of human GBM and their use in the assessment of immunotherapies [68,69].

### 4.1. GBM-Related Challenges

#### 4.1.1. Neuroanatomical Barriers

The BBB only allows small and lipophilic molecules to passively diffuse across, whereas other molecules must cross the BBB via transport mechanisms [70]. The brain parenchyma lacks conventional lymphatic vessels and the entry and activation of immune cells to the CNS is tightly regulated by the BBB to limit potential neuroinflammation. Recruitment of T cells into the brain is a coordinated process, starting with integrin adhesion molecule interactions between activated T cells and the endothelial cells. In addition to that, T cells rely on the secretion of several pro-inflammatory factors, such as tumor necrosis factor alpha (TNF-α), IL-5, and IL-12, to enter the brain parenchyma [71].

The BBTB, on the other hand, is heterogeneously disrupted and shows an increased permeability to circulating immune cells [1,72]. Furthermore, the endothelial compartment in GBM exhibits increased expression of cell adhesion molecules, which facilitates the infiltration of T cells into the brain parenchyma [71]. However, GBM also secretes chemokines like CCL17 and CCL22 that enhance the binding of immunosuppressive regulatory T (Treg) cells to endothelial cells. Therefore, GBM simultaneously creates an activated and a suppressed immune response [8]. Trafficking into solid tumors by CAR-T cells is hampered by an insufficient expression of adhesion molecules on CAR-T cells as well as a mismatch of chemokine expression at the tumor site and chemokine receptors on the surface of CAR-T cells [73]. Additionally, while the BBTB is usually compromised in the main tumor mass, it may be intact in distal brain areas infiltrated by small numbers of glioma cells [74]

#### 4.1.2. Tumor Microenvironment and Immunosuppression

In addition to the physical barriers, GBM exhibit a highly immunosuppressive milieu that supports tumor survival, growth, and infiltration and contains different immunosuppressive cells such as Tregs, MDSCs, and stromal cells. These cells are located in a remodeled extracellular matrix and are able to produce high levels of immunosuppressive cytokines and chemokines [8,36,75,76]. Furthermore, GBM express immune checkpoint ligands like programmed cell death ligand 1 (PD-L1) and induce the expression of inhibitory molecules on the surface of CAR T cells, such as programmed cell death protein 1 (PD-1) and cytotoxic T-lymphocyte-associated protein 4 (CTLA-4) [36]. Ultimately, the immunosuppressive properties of the TME inhibit T cell activation as well as their effector functions [7,8].

Tregs are immunosuppressive T cells which negatively modulate the immune response and, therefore, prevent autoimmunity. They are actively recruited by GBM tumor cells and constitute up to 30% of infiltrating lymphocytes in GBM [8,75]. GBM also recruits and polarizes macrophages toward an M2-like immunosuppressive phenotype. The M2 phenotype is characterized by a lower MHC-class II and co-stimulatory molecule expression, which has a negative impact on T cell-mediated recognition and lysis of GBM cells. Additionally, tumor-associated macrophages (TAMs) and microglia supply angiogenic molecules and growth factors, supporting GBM vascularization [8,77]. Several cell types in the GBM microenvironment, including TAMs, secrete different mediators such as cytokines (e.g., IL-6, IL-10 and TGF-ß), prostaglandins (e.g., PGE2), and chemokines (e.g., CCL20 and CXCL8) [78]. Cytokines like TGF-ß have an immunosuppressive effect, including the inhibition of T cell proliferation and the reduction of MHC class II expression on glioma cells, microglia, and macrophages. IL-10 facilitates Treg expansion and inhibits T cell effector functions [66]. Chemokines like CCL2, CCL5, and CXCL12 promote the recruitment of Tregs, TAMs, and MDSCs to GBM. Therefore, these mediators are associated with shorter OS [14,22].

Under physiological conditions, the PD-1/PD-L1 axis is important for immune homeostasis and the prevention of autoimmune responses. However, PD-L1 is upregulated in TAMs and GBM cells, which leads to immune cell dysfunction through inhibition of T cells after binding PD-1 [79]. An increased expression of inhibitory ligands is, therefore, associated with worse patient outcomes due to immune escape of the tumor [80,81]. CLTA-4 is another inhibitory immune checkpoint expressed on T cells that binds CD80 and CD86 expressed by APCs. As a result, CD28, a co-stimulatory receptor on T cells, cannot bind these ligands and contribute to T cell activation. In addition, CAR-T cell exposure to chronic antigen stimulation leads to further upregulation of PD-1 and PD-L1 [82,83], resulting in hyporesponsive T cells and enhancement of immunosuppressive Treg function [1].

The T cell motility and effector functions are a highly energy-demanding process and, therefore, modulated by the availability of several nutrients, such as glucose, glutamine, tryptophane, and arginine [83]. The high metabolic activity of the tumor cells reduces glucose and oxygen availability and causes the accumulation of lactic acid and other toxic metabolites in the TME. Consequently, T cell metabolism is constrained and the cells are nutrient deprived. Additionally, T cells suffer from oxidative stress due to hypoxia [1,24,83].

Altogether, primed T cells might infiltrate the TME of GBM, but experience inactivation or exhaustion which disables functionality, proliferative potency, and cytokine secretion and, consequently, lytic capability [20,84]. Furthermore, the immunosuppressive properties of GBM are not restricted to the TME. Patients with GBM show a state of global T cell impairment and functional immunosuppression. While systemic CD4+ T cell levels are reduced in GBM patients, inhibitory receptors such as CTLA-4, CD73, and CD39 are upregulated on peripheral T cells [66].

#### 4.1.3. Heterogeneity and Antigen Loss

A high mutational burden generates a large number of neoantigens which can offer great potential for CAR-T cell therapy against tumor-specific antigens (TSAs). However, GBM exhibit a low mutational burden [85]. Therefore, the efficacy of therapies and intrinsic immunogenicity in GBM is limited by the low neoantigen load, that could be recognized by T cells [4,75]. Simultaneously, GBM is a highly heterogeneous tumor. Typical targets for CAR-T cell therapy in GBM, such as IL-13Rα2 and EGFRvIII, are heterogeneously expressed at both the inter- and intrapatient levels and can fluctuate in a spatial and temporal manner [75]. At the cellular level, multiple subtypes and subclones reside in the same tumor with the ability to transition from one cell state to another [1,67]. This functional and molecular heterogeneity of GBM is most likely related to both clonal evolution and the plasticity of CSCs [5]. A major determinator of spatial heterogeneity are hypoxia gradients. Inside the central necrosis, microglia and macrophages are associated with an upregulation of pro-inflammatory markers while, in the periphery, more anti-inflammatory and pro-angiogenic markers are expressed [86]. There is also marked heterogeneity between primary and recurrent tumors. Interestingly, local recurrence of GBM is associated with a high stability of initial tumor mutations, while spatially distant recurrent tumors retain fewer mutations present in the initial tumor [87]. Subclonal mutations may result in a lower drug sensitivity after recurrence [1].

The heterogeneity of antigen expression in GBM impedes the potential of an antigen-specific approach such as CAR-T cell therapy. After an initial reduction of the tumor by CAR-T cells, the antigen-negative tumor cells are able to survive and evade the therapy. Consequently, the loss of targeted antigens leads to tumor relapse [19,21]. This process, in which the tumor escapes CAR targeting by downregulating or mutating the specific antigen, is called antigen escape [16]. This phenomenon occurred in different clinical trials testing CAR-T cell therapy in GBM: in one trial targeting EGFRvIII, loss of target expression and upregulation of immunosuppressive molecules were observed [60]. In another trial targeting IL-13Rα2 with specific CAR-T cells, the patient developed a recurrent tumor with decreased antigen expression [57].

Tumor heterogeneity and the probability of antigen escape result in the need for targeting multiple antigens or using combinatorial approaches [16]. On the other hand, the combination of therapies might increase the risk of toxicities, drug–drug interactions, and adverse effects [67].

### 4.2. CAR-T Cell Toxicity

Treatment with CAR-T cells is a relatively new approach to fight cancer that has been affected by several drawbacks concerning safety, toxicity, and manufacturing with no long-term experience [17].

While CAR-T cells are, theoretically, more precise and potentially less toxic than conventional systemic chemotherapy, several serious adverse events are possible during therapy. These include CRS, immune effector cell-associated neurotoxicity syndrome (ICANS), on-target/off-tumor toxicity, as well as acute respiratory distress syndrome and, in rare cases, hemophagocytotic lymphohistocytosis and macrophage activation syndrome, all of which are potentially fatal [82,88]. The major CAR-T cell toxicities can be divided into two categories, depending on the underlying mechanism: (i) general toxicities related to T cell activation and systemic release of high levels of cytokines, and (ii) on-/off-tumor toxicity, which is directed against normal tissue bearing the targeted antigen [25].

CRS develops in patients if T cell and tumor cell interactions lead to the release of a massive number of cytokines, such as interferon-gamma (INF-γ), TNF-α, IL-6, IL-10, and IL-15. Consequently, macrophages and monocytes are activated and induce a pro-inflammatory cascade of cytokines and rapid progression of CRS [20]. Patients develop a sepsis-like state that covers mild symptoms such as fatigue, nausea, headache, and fever as well as serious symptoms like rigors, hypotension, tachycardia, hypoxia, and capillary leak. In severe cases, this can lead to cardiac dysfunction, respiratory failure, hepatic failure, or disseminated intravascular coagulation with a potentially fatal outcome [18,20]. Factors that favor severe CRS include high tumor burden, high bone marrow involvement, high baseline inflammatory state, the use of high-intensity lymphodepletion with cyclophosphamide and fludarabine, as well as higher CAR-T cell dose and the type of co-stimulatory domain (e.g., CD28 carries a higher risk compared to 4-1BB) [20]. The severity and time course of CRS also depends on the tumor type and patient co-morbidities [17]. In pediatric and adolescent pre-B cell ALL, CD-19-directed CAR-T cells with a 4-IBB co-stimulatory receptor led to CRS in all 30 patients. A total of 22/30 developed mild to moderate CRS, 8/30 (27%) severe CRS, and 13/30 (43%) suffered from transient neurotoxicity that self-resolved in all patients [89]. Among various trials, the rates of CRS are around 25% [19]. The management of CRS requires a personalized approach, where mild to moderate CRS symptoms spontaneously resolve with supportive care, while, in severe cases, the IL-6 blocking agent tocilizumab is indicated [17,20].

ICANS is the second most common adverse event. In days and weeks after CAR-T infusion, neurological deficits are often seen, including cerebral edema, somnolence, agitated delirium, seizure activity, focal deficits, motor weakness, aphasia, vision changes, and tremor. The severity of ICANS appears to correlate with higher tumor burden and a more severe CRS [17,20,90]. Usually, neurotoxicity is fully reversible but, in a small number of patients, progressive cerebral edema can lead to death [16]. The exact mechanism of ICANs is still poorly understood, but it is thought that the BBB gets disrupted due to endothelial injury related to cytokine release. As a consequence, cytokines and immune cells are able to accumulate in the CNS [19,25,88].

It remains unclear whether systemic toxicities will play a role in the treatment of GBM with CAR-T cells, since GBM does not manifest with a high systemic tumor burden [2]. Solid tumors show a higher risk of on-target/off-tumor toxicity since TSAs are rare and CARs targeting TAA have the potential to attack vital tissues expressing the same antigen [91].

## 5. Outlook

Clinical trials were able to show that CAR-T cells can be applied in GBM patients with a transient response and acceptable adverse effects. To achieve a better clinical response, several improvements have been introduced in CAR-T cell technology and application mode which may improve their efficacy against GBM in the future. Current challenges and potential solutions are summarized in Table 1.

### 5.1. Breaking down Neuroanatomical Barriers

Several chemokine ligands are expressed by the tumor: CCL17 and CCL22 promote recruitment of CCR4+ Tregs, while CXCR3—along with its ligands, CXCL9 and CXCL10—drives cytotoxic lymphocyte trafficking into the GBM tumor site. To enhance T cell localization to tumors, corresponding chemokine receptors can be expressed by CAR-T cells [25,71,92]. For example, CD70-specific CAR-T cells expressing CXCR1 and CXCR2 showed enhanced migration and antitumor efficacy in murine models of GBM, ovarian, and pancreatic tumors [93]. RTx can also increase the production of chemoattractants (e.g., CXCL9, CXCL10 and CXCL11) and, thereby, promote the extravasation and expansion of T cells [94].

Another strategy is to infuse the CAR-T cell directly to the tumor to avoid the difficulty of recruitment from the blood circulation. Moreover, local infusion might also restrict on-target/off-tumor toxicities and other systemic adverse events [25,71].

As described above, three clinical trials with IL-13Rα2-CARs tested regional delivery via intralesional or intrathecal infusion in GBM patients. They were able to show tumor regression and CAR-T cell persistence at the site of intralesional infusion, as well as at sites of progression, which demonstrates an ability to traffic after local infusion. No fatal or life-threatening toxicities were observed, but fever and tachycardia were common adverse events, which might represent systemic effects despite localized delivery (reviewed in [95]). Locoregional delivery of CAR-T cells was not only tested in adults but also in children with CNS tumors [96,97,98]. On the other hand, the intracranial approach requires an invasive procedure and can be associated with certain complications (e.g., infection and bleeding). In contrast, the intravenous approach is less invasive, and intravenously infused CAR-T cells have shown the ability to cross the BBB and reach the targeted glioma tissue [63]. Thus, the optimal route for CAR-T cell delivery needs to be further elucidated and, currently, intravenous, intratumoral, and intraventricular routes are being explored [24].

### 5.2. Adjusting the TME

After migrating to the tumor site, CAR-T cells are confronted with a highly immunosuppressive microenvironment. Key molecules in the TME, such as IL-6 and TGF-ß, which contribute to immunosuppression, could be targets for monoclonal antibody therapies administered together with CAR-T therapy [14]. Moreover, CAR-T cells themselves could be programmed to express antibodies, adjuvants, cytotoxic molecules, or cytokines upon activation [16]. Secretion of cytokines such as IL-12, IL-15, IL-18, and IL-21 by so-called TRUCKs can help to promote the cells’ proliferative activity and reshape the TME, in addition to the attraction of bystanding anti-tumor immune cells [16,99]. The expression of IL-15, for instance, improved the persistence and proliferative capacity of IL-13Rα2-CAR-T cells in a preclinical glioma model and, thus, led to a stronger anti-tumor activity [32]. In addition to the secretion of cytokines, CAR-T cells can also be engineered to secrete antibody-like proteins. This may enhance CAR-T cell recognition of TAAs together with separate immune killing mechanisms mediated by the antibodies, e.g., antibody-dependent cell-mediated cytotoxicity [36].

Another strategy to improve CAR-T cell survival in cancer immunotherapy is targeting immune checkpoints, such as CTLA-4 and PD-1. To date, different approaches have been used to target immune checkpoints: (i) blocking of co-inhibitory molecules with monoclonal antibodies directed against PD-1 or PD-L1; (ii) knocking out of co-inhibitory molecules using gene-editing technologies; and (iii) expression of PD-1 switch receptors [100,101]. The targeting of co-inhibitory molecules by anti-PD1/PD-L1 monoclonal antibodies can restore cytokine production and promote CAR-T cell survival [81,102]. Yin et al. compared different approaches of checkpoint blockade in IL-13Rα2- and EGFRvIII-CAR-T cells. They showed that checkpoint blockade reversed anergy in CAR-T cells in treating murine and canine gliomas and led to an efficient tumor growth reduction. EGFRvIII CAR-T cells benefited from PD-1 and TIM-3 blockade, while IL-13Rα2 CAR-T cell efficacy was improved by CTLA-4 blockade [103]. The administration of checkpoint inhibitors together with CAR-T cells in GBM is further reviewed in [104]. CAR-T cells that secrete PD-1-blocking scFV into the local environment to block PD-1 on CAR-T cells and bystander tumor-specific T cells outperformed CAR-T cells that were combined with PD-1 blocking monoclonal antibody treatment, highlighting the role of the localized application of PD-1 blockers [105]. To overcome T cell exhaustion through checkpoint inhibition, CAR constructs have further been altered by deletion of PD-1 with the help of the clustered regularly interspaced short palindromic repeats-CRISPR-associated system 9 (CRISPR-Cas9). This led to a CAR-T cell phenotype that was less prone to exhaustion which improved the anti-tumor efficacy in murine GBM models [16,18,106,107,108]. The inhibitory intracellular signaling domain of PD-1 can also be exchanged with a stimulating intracellular signaling domain, e.g., derived from CD28. These so-called PD-1 switch receptors receive an activating co-stimulatory signal when stimulated by PD-L1 [24,25,81,109]. A throwback to this approach might be that unleashing CAR-T cells with no feedback and attenuation mechanisms could augment the risk of toxicity due to uncontrolled activation [16,18]. Similar strategies against other T cell inhibitory receptors, such as CTLA-4, TIM-3, and LAG-3, are being tested [92]. Downregulation of PD-1, TIM-3, and LAG-3 led to a fast and continuous activation of CAR-T cells on the one hand, but to reduced pool of memory cells with the possibility of a shorter time period of treatment on the other hand [94]. Further research is necessary to determine the benefit of blocking immune checkpoints [92,94].

### 5.3. Conquer Heterogeneity and Antigen Loss

As mentioned above, intra- and intertumoral heterogeneity and antigen loss might be the biggest hurdles in CAR-T cell therapy of GBM [21]. To overcome these obstacles, CAR-T cells, that can target multiple TAAs or that can flexibly target different antigens, are being developed. In particular, bi-specific or even trivalent CARs, as well as tandem CARs, can be used to target multiple antigens simultaneously. Their practicability has already been demonstrated in preclinical trials, as described above [31,35].

Another approach is to use multiple-targeted and programmable CAR-T cells, so-called Smart CAR-T cells, including SynNotch CARs, universal CARs, and split, universal, and programmable (SUPRA) CARs. This new generation of CARs contains a flexible receptor that can target various tumor antigens concurrently with high efficacy, low toxicity, and in a controllable manner [110]. Universal CARs are activated through the binding of the extracellular adaptor domain to a soluble antigen-targeting ligand, such as mABs, which are bound to the targeted antigens on the tumor cell surface. Thereby, multiple tumor antigens can be targeted selectively and flexibly, depending on the choice of the soluble ligand [110]. SUPRA CARs contain a scFv adaptor molecule and a universal receptor for flexible targeting of multiple antigens, without the requirement of complex reengineering. Thus, the treatment could be adapted to the patient’s tumor according to its changing antigen expression [110,111].

Polytherapeutic approaches could eliminate treatment resistance driving tumorigenic populations and may overcome the challenges that arise with tumor heterogeneity [87]. Next to the combination with other immunotherapeutic modalities, combinations with classical cancer therapies, such as RTx or chemotherapy, are also possible. RTx of the CNS has been shown to synergize with immunotherapy. After application of RTx, a dramatic upregulation of (neo-)antigens on the tumor cell surface can be seen, which increases the density of targets for T cells [22,114]. As an example, NKG2D CAR-T cells combined with intracranial RTx significantly reduced the tumor burden and improved survival due to the upregulation of NKG2DL after RTx [46]. Moreover, RTx leads to the release of damage-associated molecular patterns (DAMPs) at the tumor site. DAMPs, in turn, increase the expression of chemokines and, therefore, enhance the infiltration of CAR-T cells into the tumor [22,114]. Furthermore, chemotherapeutic agents such as cyclophosphamide and doxorubicin can increase the infiltration of T cells into solid tumors and improve antigen presentation [115].

### 5.4. Targeting Cancer Stem Cells

Glioma stem-like cells (GSC) are hypothesized to be one cause of treatment failure in GBM due to their higher degree of drug and radiation resistance and ability to support tumor relapse [75]. Previous studies in glioma-bearing mice have shown that the targeting of GSCs could effectively inhibit glioblastoma tumorigenesis and prolong survival. An explored target antigen is CD133, due to its association with the GSC phenotype [45,112]. The elimination of GSCs is crucial for a curative approach since these cells appear to be responsible for disease recurrence and metastatic spread [113]. Future approaches, therefore, should combine anti-GSC drugs with other therapeutic strategies [8].

### 5.5. Management of Toxicity

CAR-T cell therapy is associated with different forms of toxicity, e.g., CRS, which can ultimately prove fatal. For improving safety, different approaches have been developed: suicide strategies, synthetic splitting receptors, combinatorial target-antigen recognition, synNotch receptors, along with BiTes, and inhibitory CARs (iCAR) [99]. Control switches like off-switches or suicide genes can be used to deactivate CAR-T cells with small molecules or chemotherapeutic agents if toxic side effects take place. The downside of this strategy is that the patient loses their “living drug” that provides continuous surveillance for years after remission has been achieved [17,116]. Alternatively, the CAR gene can be placed under the control of inducible expression systems via doxycycline and tetracycline. The major limitations of this approach are the slow induction and probability of reduced receptor expression in CAR-T cells [16]. In order to achieve a more flexible “on-switch” mechanism, universal CARs can be used. Universal CARs rely on a co-administration of a soluble antigen-binding, which not only enhances safety but also the flexibility of the therapy [16]. Combinational antigen sensing is an approach to enhance the specificity and safety of CAR-T cells. In this approach, CAR-T cells require multiple TAAs to be engaged for a full activation, e.g., dual-CARs or CARs with a synNotch system. Slow activation kinetics might limit the efficacy of this approach [16]. Last, T cell activity can be restricted to the tumor tissue by using iCARs. iCARs contain an inhibitory domain derived from immune checkpoint proteins which inhibit the binding to an antigen on non-malignant cells, but not on tumor cells. [16,25,38]. Another possibility to induce spatial control is to add the hypoxia-inducible factor 1 alpha degradation pathway in order to restrict CAR expression only to those CAR-T cells that infiltrate the hypoxic TME [25,94].

## 6. Conclusions

To summarize, transient responses could be achieved in the first clinical trials of CAR-T cell therapy in GBM patients. However, the majority of patients with GBM who received CAR-T cell therapy did not experience a durable response and clinical benefit. Therefore, the clinical indication for CAR-T cells in GBM is presently undefined. Several challenges impede the success of immunotherapy in solid tumors like GBM, such as intra- and intertumoral heterogeneity, antigen loss, and an immunosuppressive TME. The upcoming generation of CAR technology will be equipped with further abilities regarding efficient antigen targeting and flexible usage of CAR-T cells. These improvements have the potential to render CAR-T cells a suitable option in GBM therapy. Further small-scale early-phase clinical trials are needed to explore the suitability, efficacy, and safety of different CAR-T cell constructs as well as their optimal route of application.

## Figures and Tables

**Figure 1 cells-12-01770-f001:**
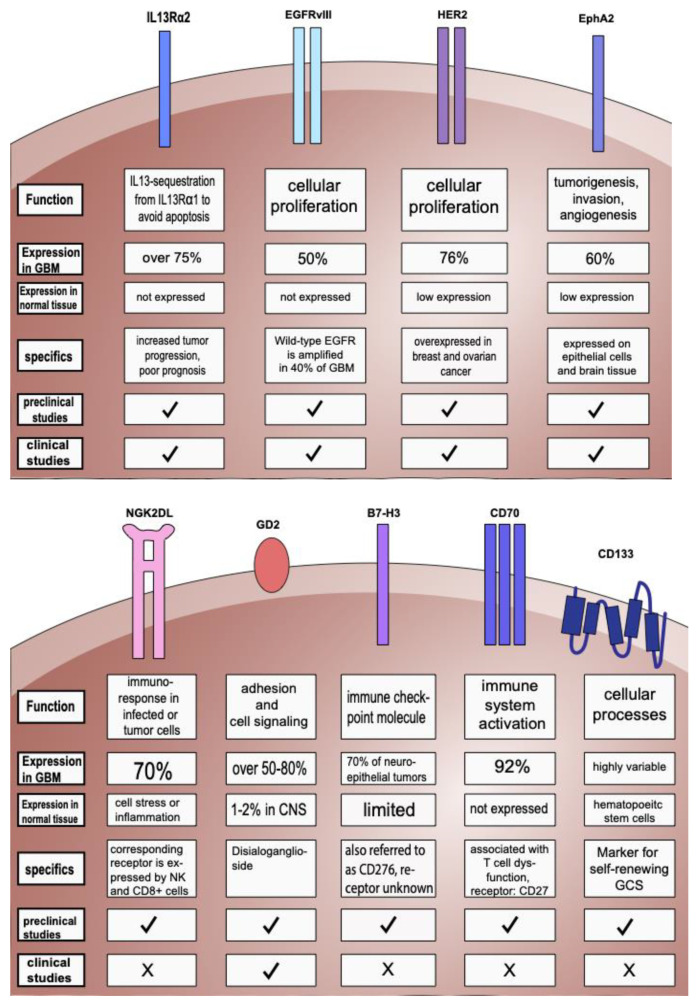
Targets used in preclinical trials and in clinical studies and their characteristics. Graphical representation of the target structures, which are all expressed on the cell surface and distinguished between expression on GBM tumor cells (at the protein level) versus normal tissue. ✓: preclinical or clinical studies were already conducted, X: no preclinical or clinical study was conducted yet. For further information regarding the specifics of target structures and references, see Appendix A.

**Table 1 cells-12-01770-t001:** Summary of current challenges and potential solutions.

Current Challenges	Potential Solutions
Limitations of GBM mouse models [68,69]	Patient-derived xenograft models in humanized mice to mimic the human immune system and intratumoral heterogeneity [68]
Neuroanatomical barriers such as the BBB and insufficient trafficking of CAR-T cells in solid tumors [1,8,70,71,72,73,74]	Chemokine receptor expression in CAR-T cells [25,71,92,93,94] Locoregional delivery of CAR-T cells [25,71,95,96,97,98]
Immunosuppressive TME [7,8,14,22,36,66,75,76,77,78,79,80,81,82,83]	CAR-T cells that secrete cytokines or antibodies that lead to antibody-dependent cell-mediated cytotoxicity [16,32,36,99] Combination with checkpoint inhibitors or secretion of checkpoint-blocking antibodies by the CAR-T cells [81,100,101,102,103,104,105] Knock-out of co-inhibitory molecules [16,18,24,25,81,106,107,108,109]
Intra- and intertumoral heterogeneity and antigen loss [1,4,16,19,21,57,60,67,75,85,86,87],	Multiple targeted and programmable CAR-T cells (synNotch CARs, universal CARs, SUPRA-CARs) [31,35,110,111] Targeting cancer stem cells to avoid recurrence with antigen loss [8,45,112,113] Polytherapeutic approaches with RTx or chemotherapy [22,46,87,114,115]
CAR-T cell toxicity [2,16,17,18,19,20,25,82,88,89,90,91]	Off-switches or suicide genes [17,99,116] Inducible expression system [16] Universal CARs [16] Inhibitory CARs [16,25,38] Combinational antigen sensing (e.g., synNotch CARs) [16,99]

## Data Availability

No new data were created or analyzed in this study. Data sharing is not applicable to this article.

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
