# Peer review of "Chimeric Antigen Receptor T Cells in Glioblastoma—Current Concepts and Promising Future"

_cells, 2023, doi:10.3390/cells12131770_

Round 1

Reviewer 1 Report

This is a very well organized and written review. 

Throughout the review it would be helpful to know what models, cell lines and PDXs are used in the cited studies.

Author Response

Reviewer 1

Comment 1: This is a very well organized and written review.

Throughout the review it would be helpful to know what models, cell lines and PDXs are used in the cited studies.

Author response: We thank the reviewer for the positive feedback. We have addressed the reviewer’s comment by adding information about the used mouse models and cell lines in the preclinical studies.

Change to text: A third generation EGFRvIII CAR-T cell with a CD28 and a 4-1BB co-stimulatory domain was evaluated in a syngeneic, fully immune-competent orthotopic mouse model after implantation of tumor cells derived of the murine glioma parental cell line SMA560vIII. In this approach, intravenous (i.v). CAR-T cell infusion with elevated doses was able to cure all mice. Cured mice were resistant to rechallenge with EGFRvIII negative tumors, suggesting that host immunity against additional antigens could have been generated [33].

An orthotopic mouse model bearing patient derived xenograft (PDX) GBM was treated with a single i.v. infusion of EGFRvIII synNotch CAR-T cells and experienced a complete and long-term remission, as well as an increased survival compared to constitutive CARs [34].

Intravenously. applied CAR-T cells, that were designed to target an epitope of EGFRvIII and a form of EGFR expressed on tumor cells but not on EGFR-expressing normal cells, were able to effectively lyse EGFRvIII- or EGFR-overexpressing tumor cells in a subcutaneous xenograft mouse model injected with EGFR-expressing tumor cells of the cell lines U87 and U251.

Intraventricular injection with CAR-T-EGFRvIII.BiTE-EGFR cells not only induced complete and durable responses associated with prolonged survival in an orthotopic mouse model, that received tumor cells of the human glioma cell lines U87 and U251, as well as in PDXs, but also redirected non-specific bystander T cells and Tregs, to become cytotoxic killers without on-target/off-tumor activity [37,38].

In an immunocompetent, orthotopic GBM model, in which mice received EGFRvIII positive GL261 cells intravenously, EGFRvIII CAR-T cells alone failed to control fully established tumors.

In order to improve delivery and efficacy of CAR-T cells, anti-VEGF therapy was combined with i.v. EGFRvIII CAR-T cells in immunocompetent mice bearing orthotopic GBM tumors fromCTA2 and GSC500 cell lines that were engineered to express EGFRvIII.

This approach was tested for targeting HER2 and IL-13Rα2 in an orthotopic xenogeneic GBM mouse model with use of intratumoral injections: TanCAR-T cells showed enhanced anti-tumor efficacy, and improved animal survival [30]. In addition, TanCAR-redirected T cells targeting IL-13Rα2 or EphA2 in a xenograft mouse model after subcutaneous injection of cells of U87 glioma cells, were able to elicit greater tumor regression than single CAR-T cells with the potential for reduction of antigen escape and off-target cytotoxicity [41].

An et al. studied third generation EphA2 CAR-T cells with CD28 and 4-1BB as co-stimulatory domains applied intravenously in a xenograft mouse model with subcutaneous injection of U251 glioma.

NKG2D CAR-T cells lysed NKG2DL expressing GBM cells and GSCs in vitro and eliminated xenograft tumors of the U251 and U87 cell lines in vivo, without significant treatment-related toxicity.

Another study in an orthotopic GBM mouse model implanted with GL-261 cells revealed the synergistic effect of NKG2D CAR-T cells and radiotherapy (RTx) with enhanced migration of the CAR-T cells to the tumor site after i.v. application and increased effector functions [46].

In another approach, GD2-targeting CAR-T cells with additional transgenic IL-15 expression showed an improved tumor control and survival of intravenously treated mice bearing patient derived, intracranial GBM [48]. Treatment with GD2 CAR-T cells in combination with the dual IGF1R/IR antagonist Linsitinib decreased the activation/exhaustion of GD2 CAR-T cells and increased their central memory profile and anti-tumor activity in an orthotopic xenograft mouse model, challenged with primary patient derived cell lines of GBM [47,48,49].

Reviewer 2 Report

In this review, the authors address the very important issue of CAR T cell therapy for glioblastoma patients. Particularly in patients at relapse after standard therapy by surgery followd by radiation therapy and temozolomide, the prognosis of the relapsed patient is very poor. Here CAR T cell therapy could make a major contribution to the fate of the patients in this infavorable situation.

The authors summarized the field very nicely. All up-to-date targets and relevant pre-clinical and clinical studies are mentioned. The work will be highly appreciated be people working in the field.

I have only few minor suggestions:

1) Regarding CD70 the following work should be mentioned as well, as it paves the way towards clinical application in the very near future also for GBM patients: Sauer T, et al. CD70-specific CAR T cells have potent activity against acute myeloid leukemia without HSC toxicity. Blood. 2021. 29;138(4):318-330. PMID: 34323938.

2) I strongly recommend that the authors add a Figure / Cartoon like a Graphical Abstract as a sort of summary of the entire work: Where would the see the clinical angle of CAR T cell therapy in the work flow / therapeutic algorithm? 

3) Maybe a second figure: Where a the target structures expressed? On which anatomical / histological sites? On the surface of the cell membrane? Intracellularly? And at which frequency are these targets expressed? At mRNA level ? Or at protein level?

Adding information and organizing it in this way would certainly majorly improve readibility to the quick reader and augment the quality of the work majorly.

Minor changes of English language could be executed by the managing editor team.

Author Response

Reviewer 2

Comment 1:

In this review, the authors address the very important issue of CAR T cell therapy for glioblastoma patients. Particularly in patients at relapse after standard therapy by surgery followed by radiation therapy and temozolomide, the prognosis of the relapsed patient is very poor. Here, CAR T cell therapy could make a major contribution to the fate of the patients in this infavorable situation.

The authors summarized the field very nicely. All up-to-date targets and relevant pre-clinical and clinical studies are mentioned. The work will be highly appreciated be people working in the field.

I have only few minor suggestions:

1) Regarding CD70 the following work should be mentioned as well, as it paves the way towards clinical application in the very near future also for GBM patients: Sauer T, et al. CD70-specific CAR T cells have potent activity against acute myeloid leukemia without HSC toxicity. Blood. 2021. 29;138(4):318-330. PMID: 34323938

Authors response: We thank the reviewer for the clear understanding of the relevance of our review and the supportive feedback, as well as the suggestion of the paper. We have addressed the reviewer’s comment by incorporating the suggested study to the preclinical studies discussed in the review.

Changes to text: CD70-specific CAR-T cells demonstrated a significant antitumor activity in different murine AML xenograft models [52]. In GBM mouse models, CD70 CAR-T cells were able to generate a potent anti-tumor response and improved survival in treated mice without adverse events [53,54].

Comment 2: I strongly recommend that the authors add a Figure / Cartoon like a Graphical Abstract as a sort of summary of the entire work: Where would the see the clinical angle of CAR T cell therapy in the work-flow / therapeutic algorithm?

Authors response:  We thank the reviewer for the suggestion. We added a figure that summarizes the CAR-T cell targets that have been explored so far in preclinical mouse models and in clinical trials with glioblastoma patients. In response to comment 3 we further added a graphical representation of the target structures, which are all expressed on the cell surface, and distinguished between expression on GBM tumor cells (at the protein level) versus normal neural tissue.

In contrast to hematological malignancies, such as B-cell lymphomas in which CAR-T cells have demonstrated impressive anti-tumor activity and are therefore integrated in the clinical work flow/therapeutic algorithms, the clinical indication for CAR-T cells in GBM is presently undefined, as clinical studies have not yet led to convincing durable responses. Therefore, further small scale, early-phase clinical trials are needed to explore the suitability of different targets in different molecular subgroups of GBM patients, the efficacy and safety of different CAR-T cell constructs, the optimal route of application and duration of treatment, and combinations with other types of immunotherapies such as immune checkpoint blockade.

We have addressed the reviewer’s comment by revision the summary.

Changes to text:  To summarize, transient responses could be achieved in the first clinical trials of CAR-T cell therapy in GBM patients. However, the majority of patients with GBM, that received CAR-T cell therapy, did not experience a durable response and clinical benefit. Therefore, the clinical indication for CAR-T cells in GBM is presently undefined. Several challenges impede the success of immunotherapy in solid tumors like GBM, such as intra- and intertumoral heterogeneity, antigen-loss and an immunosuppressive TME. The upcoming generation of CAR technology will be equipped with further abilities regarding efficient antigen targeting and flexible usage of CAR-T cells. These improvements have the potential to render CAR-T cells a suitable option in GBM therapy. Further small scale, early-phase clinical trials are needed to explore the suitability, efficacy and safety of different CAR-T cell constructs, as well as their optimal route of application.

Comment 3: Maybe a second figure: Where a the target structures expressed? On which anatomical / histological sites? On the surface of the cell membrane? Intracellularly? And at which frequency are these targets expressed? At mRNA level ? Or at protein level?

Adding information and organizing it in this way would certainly majorly improve readability to the quick reader and augment the quality of the work majorly.

Authors response: Please see response to comment 2.

Comment 4: Comments on the Quality of English Language

Minor changes of English language could be executed by the managing editor team.

Authors response: We carefully performed a proof-reading of the entire manuscript and corrected the language and phrasing.

Reviewer 3 Report

The article by Kringel et al. provides a brief review of current knowledge on CAR-T application in glioblastoma, both in animal models and in clinical trials. In addition, the authors provide a summary of the problems associated to CAR-T therapy in these tumors, and the possible strategies adopted to circumvent them. The article presents a well-structured, clear and concise overview of the topic, which is certainly of interest given the particularities of glioblastoma in terms of incidence and resistance to current treatments. However, other reviews on similar issues have been recently published (see e.g. doi: 10.3390/cancers15082351; doi: 10.3390/cancers15051414; doi: 10.3390/cancers15041249; doi: 10.3389/fimmu.2022.1008751, etc), so the contribution of this article to the knowledge in this area it is not clear.

Other specific comments:

-          It would be interesting that the authors specified the peculiarities of glioblastoma murine models and discussed whether they adequately recapitulate human disease and provide accurate information on CAR-T performance in this condition.

-          Specific information on CAR-T therapy in pediatric glioblastoma should be included.

-          A figure or table with an overview of the strategies proposed to solve current challenges associated to CAR-T therapy in glioblastoma would be desirable.

There are a number of typos throughout the text  that should be revised and corrected (e.g lines 236 , 375,,,)

Author Response

Reviewer 3

Comment 1: The article by Kringel et al. provides a brief review of current knowledge on CAR-T application in glioblastoma, both in animal models and in clinical trials. In addition, the authors provide a summary of the problems associated to CAR-T therapy in these tumors, and the possible strategies adopted to circumvent them. The article presents a well-structured, clear and concise overview of the topic, which is certainly of interest given the particularities of glioblastoma in terms of incidence and resistance to current treatments. However, other reviews on similar issues have been recently published (see e.g. doi: 10.3390/cancers15082351; doi: 10.3390/cancers15051414; doi: 10.3390/cancers15041249; doi: 10.3389/fimmu.2022.1008751, etc), so the contribution of this article to the knowledge in this area it is not clear.

Authors response: We thank the reviewer for the appreciation of our manuscript. It is true that the topic of CAR T-cells has been discussed in the glioma field before.

In their review, Wang et al. (doi:10.3390/cancers15082351) focused on the targets used in CAR-T cell therapy in GBM. They gave a detailed overview of used targets but only a short overview of challenges without the discussion of potential solution. Our review not only provides information about the described targets, but also a more detailed discussion about challenges and an outlook on newer CAR technologies as potential solutions.

Luksik et al. (doi: 10.3390/cancers15051414) described the targets used for CAR-T cell therapy in GBM, without the detailed description of preclinical and clinical studies that we provided (please see also response to comment 1 of reviewer 1). The only challenges to CAR-T cell therapy in GBM discussed in this review are antigen-loss and heterogeneity. Accordingly, the discussion about solutions is specifically about improvements on antigen targeting. Our review on the other hand, discusses a variety of challenges and potential solutions.

Pant and Lim (doi: 10.3390/cancers15041249) implemented a short overview about clinical trials with EGFRvIII, ILRalpha2 and HER2, as well as different strategies to improve the CAR-T response: bispecific CARs, resistance mechanisms to immunosuppressive cytokines in the TME, resistance to exhaustion by modifying the PD-1/PD-L1 axis and locoregional delivery. However, our review covers not only the mentioned strategies, but also includes other important topics, such as preclinical trials and a more detailed discussion of challenges.

Choi and Yin (doi: 10.3389/fimmu.2022.1008751) focused on the modifications of CARs in order to improve CAR-T cell therapy for GBM. In contrast to our review, the authors only listed the challenges of CAR-T cell therapy in GBM instead of discussing them in detail. They provide a detailed discussion of possible solutions regarding insufficient CAR-T cell response. Although we did not discuss some of the suggested solutions, we provided an overview of the most important strategies, that were mentioned in Choi and Yins review, namely: DualCARs, TanCars, BiTEs, expression of cytocines, switch-controlled CARs, chemokine receptor expression, gene editing via CRISPR to prevent T cell dysfunction and combinatorial approaches with anti-PD-1 activity.

We therefore believe that our review not only covers the most important aspects discussed in the already published reviews but provides additional information that leads to a deeper overall understanding of CAR-T cell therapy in GBM.

Comment 2: It would be interesting that the authors specified the peculiarities of glioblastoma murine models and discussed whether they adequately recapitulate human disease and provide accurate information on CAR-T performance in this condition.

Authors response: We addressed the reviewer’s comment by revision the following paragraph in the discussion of challenges of CAR-T cell therapy in GBM, as well as including potential solutions in table 2 (summary of current challenges and potential solutions). We further specified the used GBM models in preclinical study (please see response to comment 1 of reviewer 1).

Changes in text: Additionally, the species-inherent differences of GBM mouse models can lead to divergent results between preclinical and clinical studies. Main pitfalls of in vivo models of GBM include deviating biological properties and the lack of cellular heterogeneity compared to patient tumors, as well as different pharmacokinetics in animals compared to humans. Especially tumor cells derived from GBM cell lines experience a genetic drift leading to homogeneous cell populations. While PDXs can more accurately replicate human GBM regarding intratumoral heterogeneity, the standardization and reproducibility of the experimental results is limited. Furthermore, xenograft models of GBM, such as cell lines derived from human tissue and PDX models, require immunodeficient mice, limiting their ability to accurately reflect the tumor-immune interaction of human GBM and their use in the assessment of immunotherapies [68,69].

Comment 3: Specific information on CAR-T therapy in pediatric glioblastoma should be included.

Authors response: We thank the reviewer for the suggestion. There are only very few studies that address CAR-T cell therapy for GBM in a pediatric setting. We included two studies and a review of CAR-T cell therapy in pediatric GBM in the outlook.

Changes in text: Locoregional delivery of CAR-T cells was not only tested in adults, but also in children with CNS tumors [96–98].

Comment 4: A figure or table with an overview of the strategies proposed to solve current challenges associated to CAR-T therapy in glioblastoma would be desirable

Authors response:  We addressed the reviewer’s comment by incorporating a table with a summary of current challenges and potential solutions. 

Changes in text:

Table 2. Summary of current challenges and potential solutions

Current challenges

Potential solutions

Limitations of GBM mouse models [68, 69]

·       Patient derived xenograft models in humanized mice to mimick the human immune system and intratumoral heterogeneity [68]

Neuroanatomical barriers such as the BBB and insufficient trafficking of CAR-T cells in solid tumors [1, 8, 70-74]

·       Chemokine receptor expression in CAR-T cells [25,71,92-94]

·       Locoregional delivery of CAR-T cells [25,71,95-98]

Immunosuppressive TME [7,8,14,22,36,66 75-83]

·       CAR-T cells that secrete cytokines or antibodies, that lead to antibody dependent cell-mediated cytotoxicity [16,32,36,99]

·       Combination with checkpoint inhibitors or secretion of checkpoint-blocking antibodies by the CAR-T cells [81,100-105]

·       Knock-out of co-inhibitory molecules [16,18,24-25,81,106-109]

Intra- and intertumoral heterogeneity and antigen loss [1,4,16,19,21,57,60,67,75,85-87],

·       Multiple targeted and programmable CAR-T cells (synNotch CARs, universal CARs, SUPRA-CARs) [31,35,110,111]

·       Targeting cancer stem cells to avoid recurrence with antigen loss [8,45,114-115]

·       Polytherapeutic approaches with RTx or chemotherapy [22,46,87,112-113]

CAR-T cell toxicity [2,16-20,25,82,88-91]

·       Off-switches or suicide genes [17,99,116]

·       Inducible expression system [16]

·       Universal CARs [16]

·       Inhibitory CARs [16,25,38]

·       Combinational antigen sensing (e.g., synNotch CARs) [16,99]

Comment 5: There are a number of typos throughout the text that should be revised an corrected (e.g lines 236, 375,,,)

Authors response: We carefully performed a proof-reading of the entire manuscript and corrected all typos that we found.